# Long-lasting severe immune dysfunction in Ebola virus disease survivors

Aurélie Wiedemann [1], Emile Foucat[1], Hakim Hocini[1], Cécile Lefebvre [1], Boris P. Hejblum [2], Mélany Durand[2], Miriam Krüger[2], Alpha Kabinet Keita[3,4], Ahidjo Ayouba[3], Stéphane Mély [5], José-Carlos Fernandez[1], Abdoulaye Touré[3,4,6], Slim Fourati[7], Claire Lévy-Marchal[8], Hervé Raoul [5], Eric Delaporte[3], Lamine Koivogui[6], Rodolphe Thiébaut [2,9], Christine Lacabaratz[1,11], Yves Lévy [1,10,11✉] & PostEboGui Study Group*

Long-term follow up studies from Ebola virus disease (EVD) survivors (EBOV_S) are lacking. Here, we evaluate immune and gene expression profiles in 35 Guinean EBOV_S from the last West African outbreak, a median of 23 months (IQR [18–25]) after discharge from treatment center. Compared with healthy donors, EBOV_S exhibit increases of blood markers of inflammation, intestinal tissue damage, T cell and B cell activation and a depletion of circulating dendritic cells. All survivors have EBOV-specific IgG antibodies and robust and polyfunctional EBOV-specific memory T-cell responses. Deep sequencing of the genes expressed in blood reveals an enrichment in 'inflammation' and 'antiviral' pathways. Integrated analyses identify specific immune markers associated with the persistence of clinical symptoms. This study identifies a set of biological and genetic markers that could be used to define a signature of "chronic Ebola virus disease (CEVD)".

[1] INSERM U955 Team 16, Vaccine Research Institute, Faculté de Médecine, Université Paris-Est Créteil, Créteil, France. [2] Department of Public Health, Inserm Bordeaux Population Health Research Centre, Univ. Bordeaux, Inria SISTM, UMR 1219, Vaccine Research Institute (VRI), Créteil, France. [3] TransVIHMI, IRD UMI 233, INSERM U1175, Montpellier University, Montpellier, France. [4] CERFIG, Gamal Nasser University, Conakry, Guinea. [5] Laboratoire P4 Inserm-Jean Mérieux, US003 INSERM, Lyon, France. [6] Institut National de Santé Publique (INSP), Conakry, Guinea. [7] Assistance Publique-Hôpitaux de Paris, Groupe Henri-Mondor Albert-Chenevier, Laboratoire de Virologie, Créteil, France. [8] Pôle de Recherche Clinique (PRC), INSERM, Paris, France. [9] CHU Bordeaux, Bordeaux, France. [10] Assistance Publique-Hôpitaux de Paris, Groupe Henri-Mondor Albert-Chenevier, Service Immunologie Clinique, Créteil, France. [11]These authors jointly supervised this work: Christine Lacabaratz, Yves Lévy. *A list of authors and their affiliations appears at the end of the paper. ✉email: yves.levy@aphp.fr

In the 2013–2016 West African outbreak, Ebola virus infected more than 28,000 people, causing 11,310 deaths by May 11, 2016, in six countries (Sierra Leone, Liberia, Guinea, Mali, Nigeria, and Senegal)[1,2]. In the most affected countries, unprecedented follow-up of large numbers of Ebola virus disease (EVD) survivors (EBOV_S) has revealed long-term clinical sequelae. These observations raise questions about the pathophysiology of EVD, the risk of virus re-emergence and patient clinical care and treatment of EVD[3,4].

Two large cohorts of survivors from the West African outbreak have been reported[3–5]. Their clinical follow-up revealed symptom persistence >1 year after acute EVD (Prevail III cohort) or discharge from the Ebola treatment center (ETC) (Postebogui cohort). The clinical symptoms recorded during follow-up were predominantly general symptoms, musculoskeletal pain, neurocognitive, and ocular disorders.

The incidence of several new systemic symptoms was higher in survivors than in controls over 6 to 12 months of follow-up in the Prevail III cohort[4]. Symptom prevalence decreased overall, but the incidence of uveitis was significantly higher in survivors than in seronegative contacts.

In the Postebogui cohort, long-term follow-up of 802 survivors of the Guinea outbreak provided a temporal description of clinical incidence through the reporting of clinical events up to 600 days post-ETC discharge[3]. The frequencies of all clinical symptoms except ocular disorders decreased. Surprisingly, at inclusion, many patients presented symptoms similar to those experienced during acute-phase infection, suggesting the persistence or recurrence of a pathogenic process. These studies, and smaller previous case series[6–9], have refined definitions of the clinical spectrum of post-EVD sequelae.

The symptoms and clinical findings reported for EBOV_S resemble those of chronic postinfectious and/or immune dysfunction diseases. Several hypotheses concerning EVD pathogenesis have been proposed based on the persistence or increase in incidence of clinical disorders long after acute Ebola infection[10].

The immune responses of EVD patients have been reported during acute-phase infection or soon after the resolution of infection[11–17]. These studies identified immune signatures associated with death from EVD rather than survival. Fatal EVD was characterized by high inflammatory marker levels and a high viral load[12,16,17]. Conversely, survivors had significantly lower levels of inflammation[15] and robust EBOV-specific T cell responses[12]. Persistent immune activation and weak CD8+ EBOV-specific T cell responses were detectable for up to 46 days after viral clearance from plasma[13].

In this study, we took advantage of the long-term follow-up of EBOV_S, for detailed analyses of inflammatory, immune functional and phenotypic characteristics, and gene expression patterns, up to 2 years after acute EVD. Comparing these profiles with those of noninfected healthy donors (HDs), we found that EBOV_S continue to display severe abnormalities of immune function and persistent EBOV-associated immune activation.

## Results

**EBOV_S exhibit persisting symptoms and EBOV-specific immunoglobulin G (IgG).** We enrolled 35 EBOV_S, with a median age of 30 years (interquartile range (IQR): 25–36) in this study. Median [IQR] time between ETC discharge and enrollment was 23 months [19–25]. No EBOV-RNA was detectable in the blood at the time of sampling. RealStarFilovirus Screen reverse transcription polymerase chain reaction (RT-PCR) and GeneXpert assay did not detect EBOV in body fluids of these patients after inclusion in our study[3,18].

**Table 1 Characteristics of survivors and healthy donors.**

|  | HD | EBOV_S | P value |
|---|---|---|---|
| No. of participants | 39 | 35 | N/A |
| Median age (years) | 25 [21–36] | 30 [25–36] | 0.2 |
| Sex (male) | 31 (80%) | 19 (54%) | 0.03 |
| Median time from ETC discharge to inclusion (months) | N/A | 23 [19–25] | N/A |
| EBOV RT-PCR[a] | N/A | Negative (100%) |  |
| Clinical events[b] |  |  |  |
| All symptoms | N/A | 23 (66%) | N/A |
| Joint pain | N/A | 13 (37%) | N/A |
| Fatigue | N/A | 12 (34%) | N/A |
| Ocular disorders | N/A | 6 (17%) | N/A |
| Headache | N/A | 6 (17%) | N/A |
| Muscle pain | N/A | 5 (14.3%) | N/A |
| Fever | N/A | 4 (11.4%) | N/A |
| Abdominal pain | N/A | 2 (5.7%) | N/A |
| Anorexia | N/A | 1 (2.8%) | N/A |

The differences between healthy donors (HDs) and survivors (EBOV_S) were evaluated in nonparametric Mann–Whitney $U$ tests or $\chi^2$ test. Median values ± IQR are shown. Source data are provided as a Source Data file.
[a]Blood samples were taken from patients enrolled in the Postebogui cohort. Before PBMC and serum freezing, EBOV RT-PCR was performed on each sample to exclude the presence of EBOV in the blood.
[b]Symptoms and findings on physical examination for the EBOV_S on inclusion in the Postebogui cohort.

The enrolled individuals received only supportive care (no experimental drugs and no convalescent plasma) during the acute phase of EBOV infection and were seronegative for human immunodeficiency virus, hepatitis C virus, and hepatitis B virus. On inclusion in Postebogui cohort, 23 of the 35 patients (66%) had post-EVD symptoms similar to those for the whole cohort[3] (Table 1 and Supplementary Fig. 1). More precisely, we looked at the clinical signs of $n = 34$ EBOV_S, at the harvest time (visit 0, V0) of this immunological study, within 1 month (M1), and within 3 months (M3). At V0, 16/34 EBOV_S experienced at least one symptom with the following distribution: general symptoms (fatigue, anorexia, fever, pallor, abdominal pain, and pelvic pain) ($n = 14$), musculoskeletal symptoms (arthralgia and myalgia) ($n = 4$), neurological symptoms (headache, insomnia, vertigo, and sensory disorders) ($n = 10$), and ocular symptoms (conjunctivitis, ocular disorders, and ocular pain) ($n = 2$). Within M1, 18/34 experienced at least one symptom. These symptoms were general signs ($n = 16$), musculoskeletal symptoms ($n = 7$), neurological symptoms ($n = 12$), and ocular symptoms ($n = 2$). Within M3, 27/34 EBOV_S experienced at least one symptom. More precisely, $n = 23$ patients experienced a general symptom, $n = 13$ patients a musculoskeletal symptom, $n = 20$ patients a neurological symptom, and $n = 3$ patients an ocular symptom (Supplementary Fig. 2).

The median mean fluorescence intensity (MFI) [IQR] of EBOV-specific antibodies was 3341 [1365–7154], 930 [548–1621], 640 [413–919], and 1288 [649–2328] against NP, GP-Kissidougou, GP-Mayinga, and VP40, respectively. By contrast, none of the serum samples from HDs tested positive for these antibodies (Supplementary Fig. 3). Thus, 23 months after EVD infection, EBOV_S had EBOV-specific IgG.

**Chronic activation and microbial translocation in EBOV_S.** Twenty-two markers of inflammation and immune activation were evaluated using a 22-Plex Assay Kit measuring cytokine release syndrome-associated inflammatory cytokines. Five of the 22 soluble mediators quantified in the serum were present at significantly higher levels in EBOV_S than in controls. Median

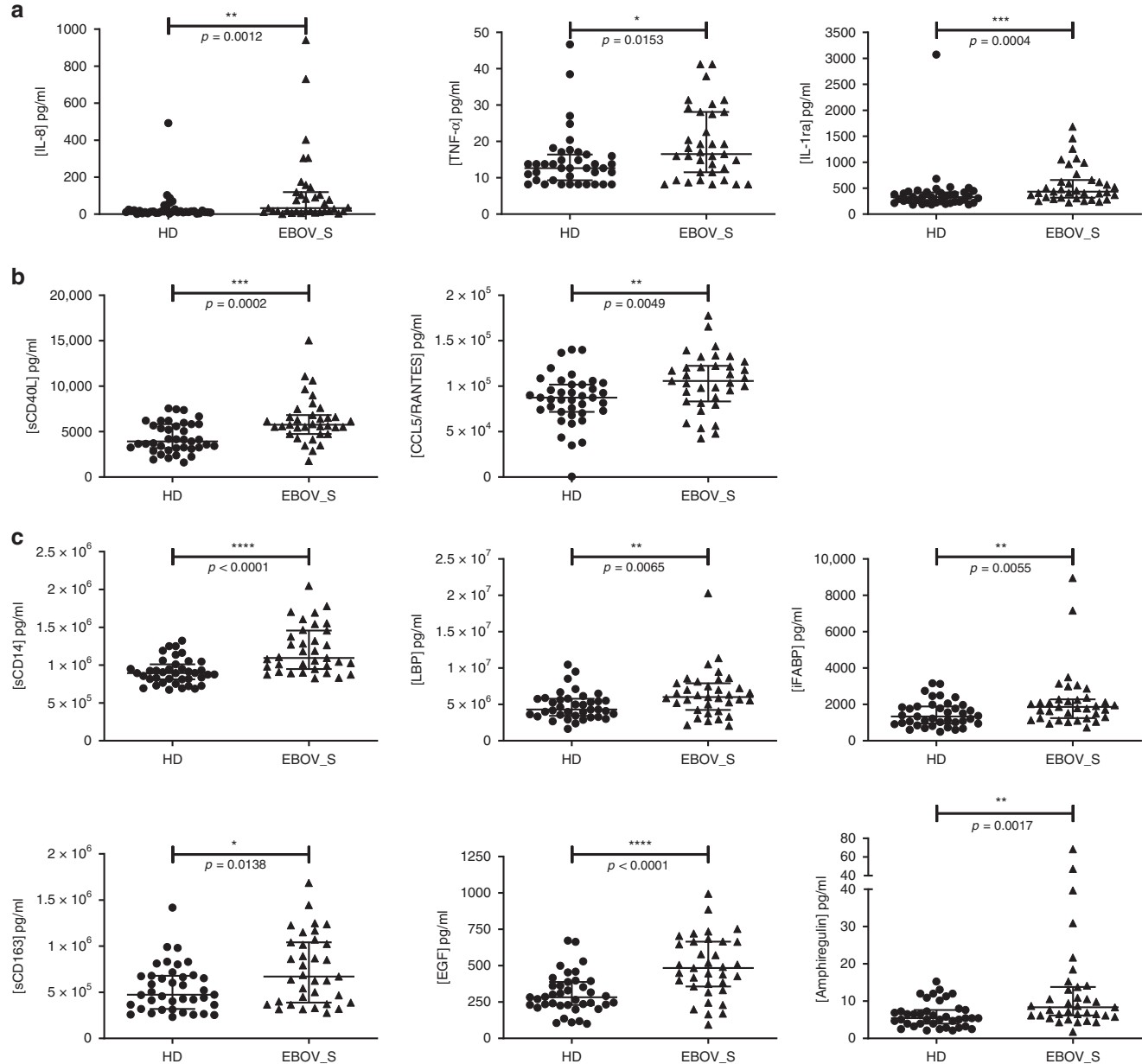

**Fig. 1 Quantification of serum-soluble mediators differentially expressed in HDs and EBOV_S.** Measurement of serum-soluble mediators (pg/ml) from $n = 39$ HDs and $n = 35$ EBOV_S with the Bio-Plex 200 System[TM] (Bio-Rad). Pro-inflammatory and anti-inflammatory cytokines (**a**). Markers of T cell function (**b**). Markers of gastric tissue integrity (**c**). The differences between HDs and EBOV_S were evaluated in nonparametric Mann–Whitney $U$ tests. Median values ± IQR are shown. Source data are provided as a Source Data file.

[IQR] pro-inflammatory cytokine concentrations were 33 pg/ml [15.9–118.9] vs. 14 pg/ml [9.8–21.6] ($p = 0.001$, for the Mann–Whitney $U$ test) for interleukin-8 (IL-8) and 16.5 pg/ml [11–28] vs. 12.6 pg/ml [9.3–16.4] for tumor necrosis factor-α (TNFα) ($p = 0.015$, for the Mann–Whitney $U$ test). For the anti-inflammatory cytokine IL-1 receptor antagonist (IL-1RA), median [IQR] concentrations were 433.8 pg/ml [315–657] vs. 327.9 pg/ml [221–419] ($p = 0.0004$) (Fig. 1a). The T cell function markers soluble CD40L (sCD40L) and CCL5 were also present at significantly higher levels in survivors than in HDs ($p = 0.0002$ and $p = 0.005$, for the Mann–Whitney $U$ tests, respectively) (Fig. 1b).

The persistence of markers of chronic activation in the blood of survivors suggested possible microbial translocation from a leaky gut, as described in other chronic infectious diseases[19,20]. Seven markers of gut tissue integrity[21,22] were also evaluated either by

Luminex technology or enzyme-linked immunosorbent assay (ELISA). Survivors had significantly higher levels of soluble CD14 ($p < 0.0001$, for the Mann–Whitney $U$ test), lipopolysaccharide-binding protein (LBP) ($p = 0.007$, for the Mann–Whitney $U$ test), intestinal fatty acid-binding protein (iFABP), a marker of gut epithelial damage ($p = 0.006$, for the Mann–Whitney $U$ test), and soluble CD163, a specific marker of monocyte/macrophage activation ($p = 0.014$, for the Mann–Whitney $U$ test) (Fig. 1c). Consistently, epidermal growth factor (EGF), a secreted gastric protection protein[23] was also present at higher levels in EBOV_S than in HDs: 483.6 pg/ml [356–665] vs. 282.6 pg/ml [230–388] ($p < 0.0001$, for the Mann–Whitney $U$ test). The EGF-like molecule, amphiregulin, was also present at higher levels in EBOV_S: 8.4 pg/ml [6.2–13.8] vs. 5.45 pg/ml [4–7.7] ($p = 0.001$, for the Mann–Whitney $U$ test) (Fig. 1c).

**Immune cell subset frequencies are changed in EBOV_S.** Cell phenotyping on blood from survivors (representative data, Supplementary Fig. 4) showed a higher frequency of total CD8$^+$ T cells among CD3$^+$ cells ($p = 0.0004$, for the Mann–Whitney $U$ test) and higher levels of activated CD8$^+$ T cells (CD8$^+$HLA-DR$^+$CD38$^+$) ($p = 0.01$, for the Mann–Whitney $U$ test) than in controls (Fig. 2a). Total CD19$^+$ B cell frequency was similar in the two groups, as was the frequency of plasmablasts (data not shown), but the frequencies of activated memory and exhausted B cells were higher in EBOV_S ($p = 0.006$ and $p < 0.0001$, for the Mann–Whitney $U$ tests, respectively) (Fig. 2b). Previous reports have shown a depletion of T and natural killer (NK) cells during EBOV infection[24,25]. We found here, 23 months after ETC discharge, a trend towards lower total NK cell frequencies in EBOV_S, and a significant downregulation of the expression of the activation marker NKG2D relative to controls ($p = 0.004$, for the Mann–Whitney $U$ test). Moreover, the frequency of non-classical CD56$^-$CD16$^+$ NK cells remained high in EBOV_S ($p < 0.0001$, for the Mann–Whitney $U$ test) (Fig. 2c). Previous studies suggested that antigen-presenting cells are initial targets to EBOV[26,27]. Blood analysis revealed a severe deficit of total dendritic cells (DCs) (HLA-DR$^+$, Lin$^-$) ($p = 0.01$, for the Mann–Whitney $U$ test), mostly due to a deficit of plasmacytoid DCs (pDCs) (HLA-DR$^+$/Lin$^-$/CD45RA$^+$/CD123$^+$) ($p = 0.006$, for the Mann–Whitney $U$ test), whereas conventional DC (cDC) levels remained in the normal range. Higher levels of expression were observed for the activation marker CD40 in both cDCs ($p = 0.03$, for the Mann–Whitney $U$ test) and pDCs ($p = 0.02$, for the Mann–Whitney $U$ test), with higher levels of HLA-ABC expression in pDCs ($p = 0.02$, for the Mann–Whitney $U$ test) (Fig. 2d). We also evaluated frequency of monocytes using CD14$^+$ and CD16$^+$ markers. Classical (CD14$^{++}$CD16$^-$) and nonclassical (CD14$^+$CD16$^{++}$) monocytes were not significantly different between the two groups (data not shown). However, pro-inflammatory intermediate monocytes (CD14$^{++}$CD16$^+$) were significantly increased in EBOV_S as compared to HDs ($p = 0.003$, for the Mann–Whitney $U$ test) (Fig. 2e).

**EBOV_S have robust EBOV-specific memory T cell responses.** Characterization of EBOV-specific T cell responses in Ebola infection are still limited and most of the studies evaluated these responses either during the acute or the recovery phase of EBOV infection comparing EBOV_S and fatal cases[11–17]. We assessed whether EBOV-specific T cells would become detectable after antigen-specific in vitro T cell expansion. A high frequency of functional CD4$^+$ and CD8$^+$ T cells producing cytokines in response to various EBOV peptides was detected in survivors. Median frequencies [IQR] of CD4$^+$ and CD8$^+$ cytokine$^+$-specific T cells were 4.89% [1.93–11.2] and 12.4% [8.9–25.9], respectively, for the EBOV1 peptide pool and 5.18% [3.4–12.6] and 11.8% [5.1–19], respectively, for the EBOV2 peptide pool ($p < 0.0001$ for all comparisons to nonstimulated conditions, for the Friedman's test) (Fig. 3a). EBOV stimulation elicited IFN-γ-positive (IFN-γ$^+$), TNF$^+$, macrophage inflammatory protein-1β-positive (MIP-1β$^+$) and IL-2$^+$ CD4$^+$ T cells ($p < 0.0001$, for each cytokine, for the Friedman's test) and IFN-γ$^+$, TNF$^+$, MIP-1β$^+$ CD8$^+$ T cells ($p < 0.0001$ for each cytokine, for the Friedman's test) (Supplementary Fig. 5a). EBOV-specific CD8$^+$ T cells expressing the cytotoxicity markers CD107a and IFN-γ after peptide stimulation were also detected at high frequency ($p < 0.0001$, for all stimulation conditions, for the Friedman's test) (Fig. 3b). A large proportion of the EBOV-specific CD4$^+$ and CD8$^+$ T cells were polyfunctional, producing up to four cytokines simultaneously (Fig. 3c). As a control, we also evaluated EBOV-specific responses in seven HDs with peripheral blood mononuclear cells (PBMCs)

available in the same conditions as used for EBOV_S. We did not detect significant specific responses against the peptide pools (EBOV1 and 2) compared to nonstimulated culture conditions. Similarly, EBOV-specific CD8$^+$ T cell expressing the cytotoxicity markers CD107a and IFN-γ were not significantly detected after peptide stimulation (Supplementary Fig. 6).

**Enrichment in inflammation and antiviral pathways in EBOV_S.** An analysis of whole-blood gene expression profiles (26 EBOV_S and 33 HDs) revealed differential expression for 1024 genes. Ingenuity Pathway analysis showed 556 annotated genes (112 upregulated and 444 downregulated in survivors) defining the cluster of survivors (Fig. 4a). The differentially expressed genes characteristic of survivors included genes involved in immune responses: IFN signaling, the complement system, acute-phase response signaling and pattern recognition receptors (PRRs) (Fig. 4b). EBOV_S displayed a clear upregulation of genes relating to the antiviral response involving IFN signaling (*IFIT1*, *IFI6*, *IFIT3*, *ISG15*, *OAS1*, *IFITM3*, and *MX1*), the complement system (*C4BPA*, *SERPING1*, *C1QC*, *C1QB*, *C1QA*, *C2*), and PRR (*C1QA*, *C1QB*, *C1QC*, *IL10*, *OAS1*, *OAS3*, *PRKD1*) signaling pathways (Fig. 4c).

**Immune markers associated with clinical symptoms.** Association between the various markers has been explored through several approaches. First, a principal component analysis (PCA) and Spearman's correlation matrix were performed to better illustrate the interrelationships between phenotypic, seric, and genetic markers. As shown in Fig. 5a, the first two components of the PCA explained 35% of the variability of the markers. Interestingly, when representing the distribution of the individuals according to their clinical status (EBOV_S or HDs), the first component separated quite well the two groups (Fig. 5b), although this analysis was unsupervised since the clinical status information was not used to construct the components. When looking at the contribution of markers over the two components, clearly, the first component separated total DCs and pDCs (projected on the left) from inflammatory markers (projected on the right). These results are consistent and extended data from the Spearman's correlation matrix (Fig. 5c) showing that inflammatory markers were positively correlated themselves and also with phenotypic markers (such as exhausted B cells, CD40$^+$DC, CD38$^+$HLA-DR$^+$CD8$^+$ activated T cells) and seric markers (such as IL-8, soluble CD14 (sCD14), and T cell activation markers CCL5 and sCD40L), for example, and inversely correlated with frequencies of blood total DC and pDC populations. A projection of the selected genes differentially expressed between EBOV_S and HDs on the same plan showed that all IFN signaling genes aggregated with the inflammatory markers, whereas the downregulated genes *C1S*, *FGFR4*, and *MASP2* (Fig. 4c) appeared on the total pDC side (Fig. 5d).

Second, we looked at the distribution of immunological markers identified above according to the presence of clinical symptoms using Wilcoxon's rank-sum statistical test (without test multiplicity correction in this explanatory analysis). At V0 time point, we found that EBOV_S with symptoms exhibited a lower frequency of blood pDCs as compared to those without symptoms (median 0.2% IQR [0.16–0.25] vs. 0.38% [0.26–0.43]; $p = 0.005$). A trend ($p = 0.06$) for an association with a higher frequency of the nonclassical CD56$^-$CD16$^+$ NK cell population (16.1% [9.32–23.12] vs. 9.06% [3.7–14.2]) was noted. These associations at V0 time point were still significant within M1, where individuals with symptoms were those with low and high frequencies of pDCs (0.2% [0.15–0.25] vs. 0.38% [0.28–0.42]; $p = 0.007$) and CD56$^-$CD16$^+$ (16.1% [9.07–23.17]) vs. 8.43% [3.4–13.8]; $p = 0.02$) populations,

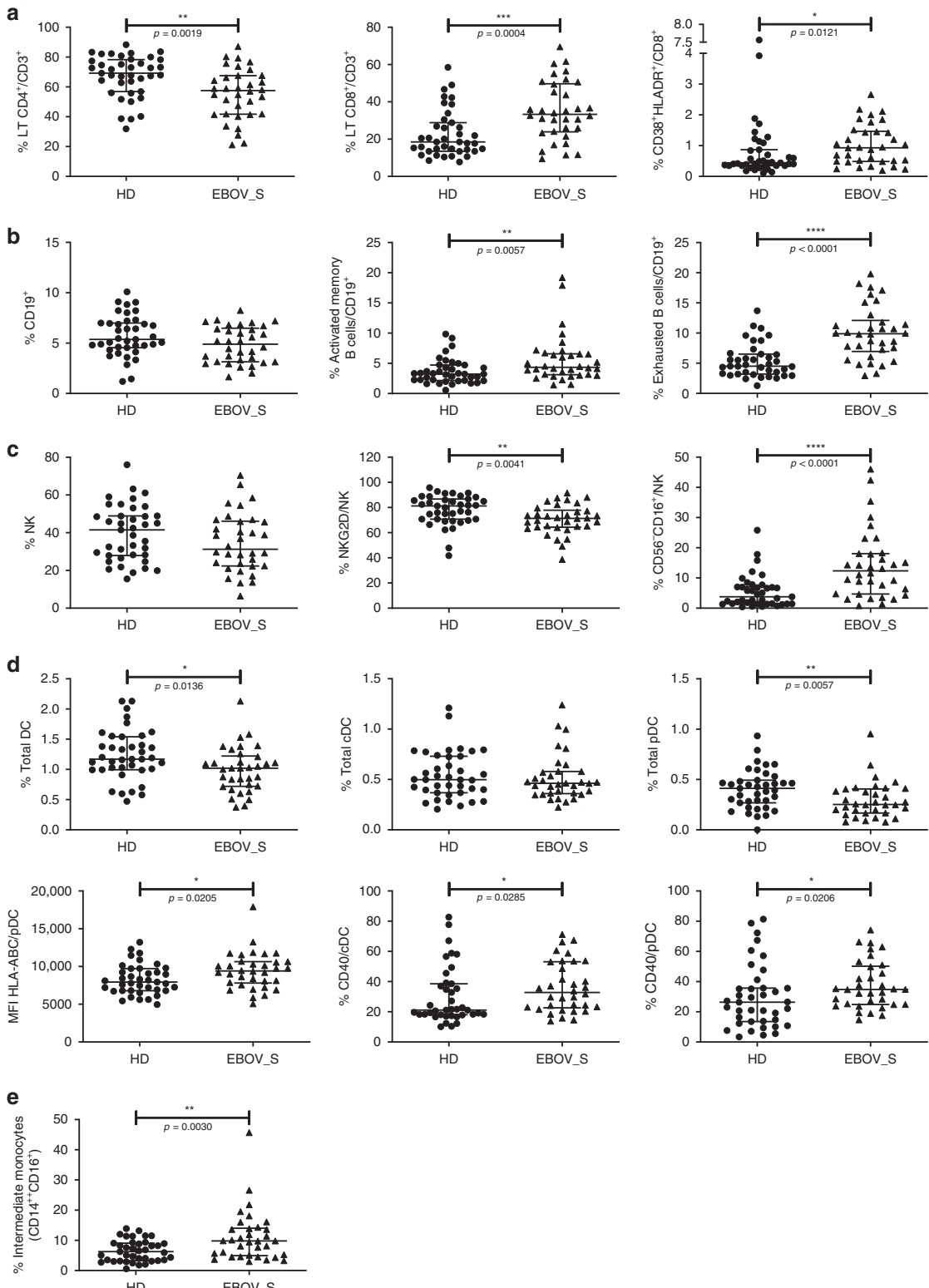

**Fig. 2 Modifications in immune cell subset frequencies in EBOV_S.** Cumulative T cell frequency and CD8 T cell activation analyses (**a**), B cell subsets (**b**), NK cell subsets (**c**), dendritic cell subsets and activation (**d**), and monocyte subsets (**e**) from $n = 34$ EBOV_S and $n = 39$ HDs. DC gating strategy is shown in Supplementary Fig. 4. The differences between HDs and EBOV_S were evaluated with nonparametric Mann–Whitney $U$ tests. Median values ± IQR are shown. Source data are provided as a Source Data file.

respectively. A trend ($p = 0.04$) toward a decrease of sCD163 (583.4 ng/ml [375.44–836.45] vs. 880.06 ng/ml [595.34–1182.4]) levels was noted in individuals with symptoms within M1. The independent association of each of these markers (i.e., pDCs, CD56⁻CD16⁺, and sCD163) with the presence of symptoms was confirmed by a multivariable logistic regression model (all $p$ values <0.02) with a reasonable goodness-of-fit statistics (Hosmer and Lemeshow test: $p = 0.11$) (Supplementary Fig. 7).

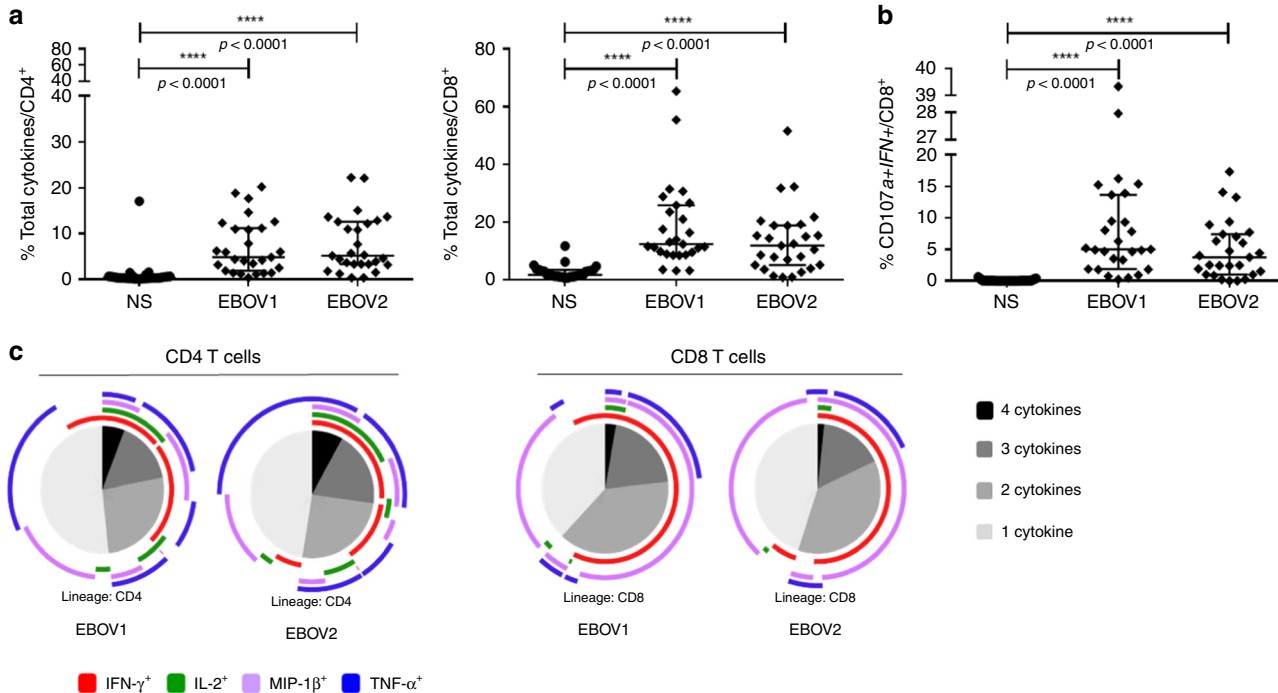

**Fig. 3 Robust and polyfunctional responses in EBOV_S.** EBOV GP-specific CD4+ T cell (left panel) and CD8+ T cell (right panel) responses of EBOV_S ($n = 27$) after 9 days of EBOV GP-specific (EBOV1 and EBOV2 peptide pools) T cell expansion in vitro (all cytokines) (**a**). Analysis of the co-expression of CD107a and IFN-γ by EBOV GP-specific CD8 T cells from EBOV_S ($n = 27$) after 9 days of antigen-specific T cell expansion in vitro (**b**). Median values ± IQR are shown, and Friedman's test was used for comparisons. Functional composition of EBOV GP-specific CD4+ and CD8+ T cell responses (**c**). Responses are color coded according to the combinations of cytokines produced. The arcs identify cytokine-producing subsets (IFN-γ, IL-2, MIP-1β, and TNF) within the CD4+ and CD8+ T cell populations. Source data are provided as a Source Data file.

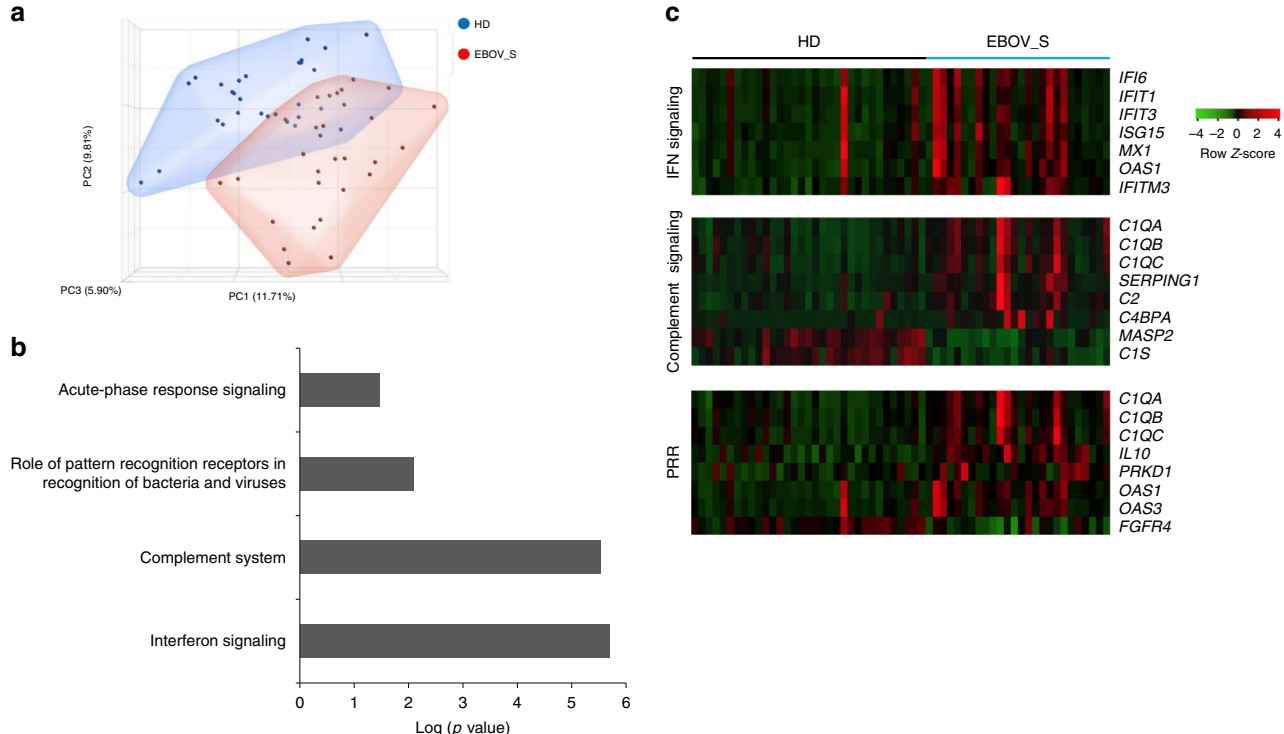

**Fig. 4 Enrichment of genes associated with inflammation and antiviral response in EBOV_S.** Unsupervised principal component analysis (PCA) of EBOV_S ($n = 26$) and HDs ($n = 33$). EBOV_S are indicated in red and HDs in blue (**a**). Ingenuity Pathway software analysis of the genes involved in immune responses differentially expressed in EBOV_S and HDs (**b**). Heatmap of genes from the main pathways associated with differentially expressed genes in EBOV_S and HDs, including IFN signaling, the complement system and PRR signaling pathways. Each column depicts one subject (HD or EBOV_S) (**c**).

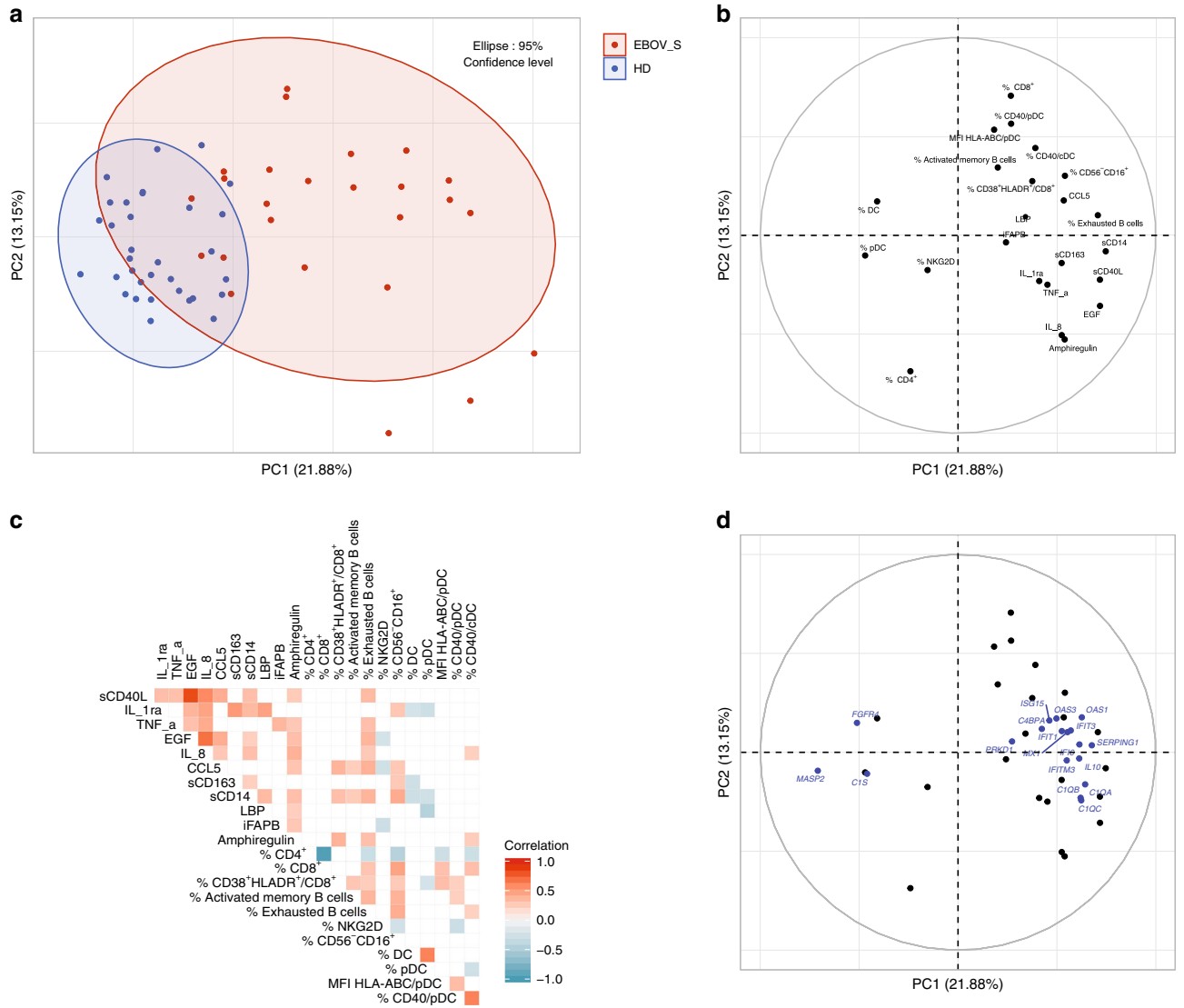

**Fig. 5 Persistence of innate immunity dysfunction in EBOV_S.** Unsupervised principal component analysis of serum-soluble mediators and cell phenotypic data, including $n = 26$ EBOV_S and $n = 33$ HDs. EBOV_S are indicated in red and HDs in blue (**a**). Variables used for the construction of the components (**b**). Spearman's correlation matrix between serum-soluble mediators and cell phenotypes from $n = 39$ HDs and $n = 34$ EBOV_S. Colors indicated Spearman's correlation coefficient. Only significant correlations ($p < 0.05$) are represented (unadjusted for test multiplicity) (**c**). Unsupervised PCA including genes (blue) from the main pathways associated with differentially expressed genes in EBOV_S and HDs, including IFN signaling, the complement system and PRR signaling pathways (**d**).

Globally, these data revealed a low frequency of pDCs contrasting with an expansion of nonclassical NK cell populations in the blood of survivors with symptoms pointing out a persistence of innate immunity dysfunction.

**Comparable immune signatures in EBOV_S, SLE, and acute EBOV.** Owing to previous published data from other groups, we also compared gene expression profiles of EBOV_S from our cohort with a chronic "noninfectious" inflammatory disease, such as systemic lupus erythematosus (SLE) and with acutely Ebola-infected individuals[28],[29]. Comparison of gene expression pathways of survivors with signature reported in SLE patients using Chaussabel's modules[28] showed that some relevant modules were similarly differentially expressed in SLE patients and EBOV_S such as M3.1 IFN pathway, M2.1 cytotoxic cells, M2.8T cells, and M2.6 myeloid lineage. Not unexpectedly, there are significant differences in other pathways, such as for example M2.4 (ribosomal proteins) (Fig. 6a, b). We also compared our results with

published data on gene expression in Ebola acutely infected individuals[29]. The full head-to-head comparison with acute EBOV patients was not possible because of more stringent criteria used to define the signature in this previous study[29]. However, this analysis showed that the IFN module M3.1, which is over-expressed in acute patients, persists in EBOV_S (list of genes from acute EBOV transcriptomic signature belonging to IFN pathway (M3.1) module, Supplementary Fig. 8).

**Discussion**
The 2013–2016 Ebola outbreak in West Africa revealed how little was known about the pathogenicity of this virus and markers predictive of its clinical outcome. Several studies have investigated the acute phase of infection in EVD patients, or the period shortly after viral clearance. By contrast, we studied long-term survivors of the recent outbreak in Guinea. By comparing these survivors with a cohort of volunteers who had not had EVD and lived in the same areas, we were able to identify a profile specific to

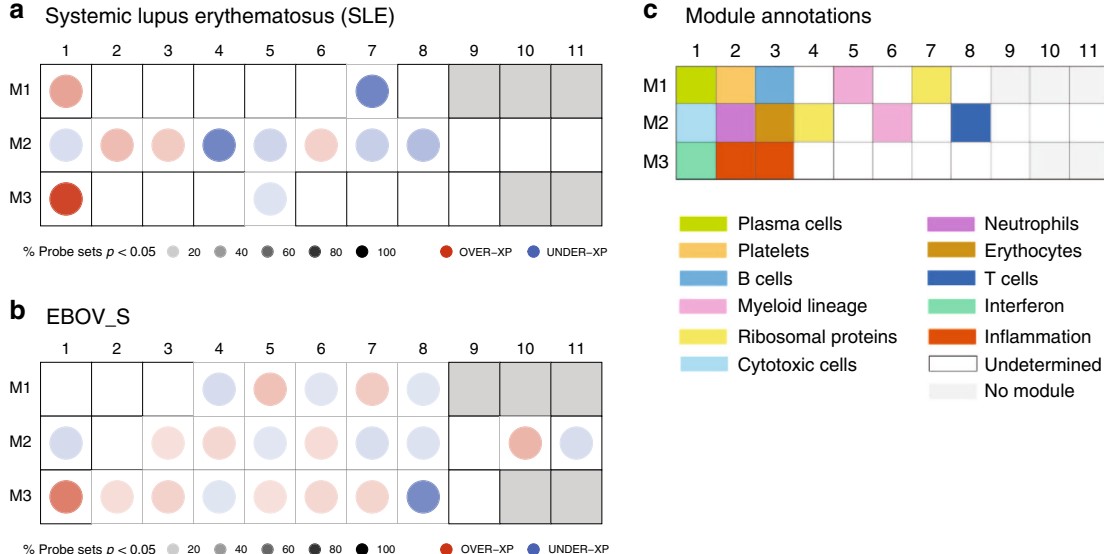

**Fig. 6 Similarities between EBOV_S and SLE and acute EBOV immune signatures.** Mapping global transcriptional changes with the use of Chaussabel's modules[28] for which at least 15% of the transcripts are significantly changed between controls ($n = 12$) and patients with SLE ($n = 22$) (**a**) or HDs ($n = 33$) and EBOV_S ($n = 26$) ($p$ value <0.05 for the Mann–Whitney/Wilcoxon's test) (**b**). Module annotations (**c**). Spots indicate the proportion of genes significantly changed for each module ($p$ value <0.05 for SLE as per Chaussabel et al.[28], fold change >1.5, and FDR <0.05 for EBOV_S). Coordinates indicate module ID (e.g., M2.8 is row M2, column 8). Overexpressed modules are indicated in red and underexpressed modules are indicated in blue, in comparison to control groups. Functional interpretation is indicated on a grid by a color code[28].

survivors. Our results, obtained with a large array of assays, highlight the existence of a consistent, intense chronic immune activation and inflammatory profile in survivors. Up to 2 years after healing and discharge from the ETC, survivors have persistently high serum levels of pro-inflammatory cytokines (IL-8 and TNF) and chronic immune activation markers (CCL5 and sCD40L). Consistent with these observations, an abnormal expression of activation markers was observed on circulating DCs, and the balance of immune cells was shifted towards circulating activated CD8+ T cells, exhausted B cells, pro-inflammatory monocytes, and nonclassical NK cells, a population reported to be abundant in other chronic viral infections[30,31] and in patients with acute EVD who subsequently survived[32].

To evaluate association between immune parameters of EBOV_S highlighted in our study and well-characterized clinical post-EVD symptoms[3,4], we performed integrated analysis that revealed a lower frequency of pDCs contrasting with an expansion of nonclassical NK cell populations in the blood of survivors with symptoms as compared to survivors without symptoms, pointing out a persistence of innate immunity dysfunction up to 2 years after healing.

Finally, deep-sequencing analyses of gene expression in the blood showed a significant enrichment in the expression of genes associated with IFN signaling, complement, pattern recognition receptor, and acute-phase response signaling. Previous studies have reported similar profiles for patients with acute EVD[14,29]. Comparison of EBOV_S immune signatures with published data on gene expression in acute EVD patients[29] showed that IFN pathway overexpressed in acute patients was persisting in survivors enrolled in our study. Furthermore, comparison of EBOV_S immune signatures with SLE, a chronic noninfectious disease, showed that IFN, T cell, cytotoxic cell, and myeloid cell pathways looked similarly differentially expressed.

The persistence of activation/inflammation pathways in EBOV_S could be due to an increase susceptibility to viral infections or reactivations of latent viruses. EBOV_S enrolled in our study were assessed for biological parameters at inclusion and during follow-up visits[3]. Longitudinal C-reactive protein measurements showed no evidence of inflammatory syndrome. Furthermore, EBOV_S exhibited negative CMV and EBV viral loads at the time of immunological study. Therefore, these clinical and biological data likely suggest no evidence of intercurrent disease, which may explain the profile of EBOV_S.

It has been suggested that survivors have a profile of weaker inflammation, and less intense cytokine and chemokine "storms" than fatal cases, in whom these abnormalities are correlated with viral replication and tend to increase until death. Our data extend these findings, by showing that the long-term persistence of these abnormalities, in the absence of detectable viral replication, may be a signature of a "chronic EVD (CEVD)," and shedding light on its pathophysiology.

Clinically, the production of a damage-associated molecular pattern during acute EVD was thought to be associated with tissue damage, leading to a pathogenic activation cascade and, ultimately, to a clinical syndrome resembling septic shock[17]. We did not perform gut biopsies on these survivors, for practical reasons, but our results show that the chronic activation profile in CEVD is associated with high blood levels of markers of intestinal permeability and microbial translocation from a leaky gut (sCD14, iFABP, LBP, and sCD163). Acute EVD is characterized by severe gastrointestinal symptoms[33,34]. Our data suggest there may be substantial long-term gut damage, resulting in structural impairment of the epithelial barrier, as reported in several other chronic infectious diseases[19,20]. Consistently, levels of EGF, which is present at higher levels during the acute phase of EVD in individuals who survive than in those who die[15], and amphiregulin, which restores tissue integrity following inflammation-associated damage[35], remained higher in EBOV_S than HDs.

Identifying the immune responses associated with protection against Ebola infection or survival is crucial. The development of strong T cell responses during acute infection has been shown to be important for viral clearance and survival, whereas B cell responses seem to be less efficient with very few potent B cell clones[12,36]. However, data remain scarce for humans. We show

here that survivors display strong, robust polyfunctional memory T cell responses to various Ebola epitopes. The antiviral functional capacity of these cells was not studied, but the phenotype of CD8[+] T cells after in vitro stimulation (strong cytotoxicity marker expression) suggested that they remained cytotoxic. We were unable to study the progression of these responses longitudinally, but the high frequency of these cells 2 years after viral clearance is intriguing. Survivors also maintained strong specific IgG responses against various Ebola proteins, confirming previous studies from precedent outbreaks[37,38]. These data and the recent demonstration of a change in IgG isotype during maturation of the anti-Ebola IgG repertoire in four survivors[39] suggest that specific anti-Ebola responses change over time, consistent with possible persistent exposure to viral proteins captured at immunological sites or localized smoldering viral replication at sites of immune privilege (testes, eyes, and central nervous system)[40–42]. The ability of these responses to protect against secondary infection or control residual replication, thereby containing viral reservoirs, is unknown, but would have major implications for vaccine development.

Ebola disease remains a public threat. The mobilization of national and international organizations, community health workers and patients, civil society, and policy makers is crucial to contain epidemic spread and decrease mortality. Survivors of the 2013–2016 epidemic in West Africa and the current outbreak in the Democratic Republic of Congo ("les vainqueurs d'Ebola") are at risk of developing a profile of severe immune dysfunction and increased morbidity that we propose to call "CEVD." Our data provide a set of biological and genetic markers for assessing clinical outcomes and highlight the importance of developing such leading-edge studies, despite limited infrastructures in an epidemic context and the need for resources for the long-term follow-up of survivors.

## Methods

**Participants**. We enrolled a subgroup of post EBOV_S from the Postebogui cohort in this ancillary immunological study. The design of the Postebogui cohort and patient characteristics have been described elsewhere[3,5,43]. Eligible patients with laboratory-confirmed EVD subsequently declared virus free were recruited at the ETCs in Guinea between March 2015 and July 2016. Ebola RNA was screened on body fluids (semen, urine, cervicovaginal secretions, breast milk, and saliva) using the standard RealStarFilovirus Screen RT-PCR Kit 1.0 and an additional in-house technique targeting the viral nucleoprotein (NP)[3]. Ebola Xpert assay was also used to re-evaluate, retrospectively on available left-over samples, RNA detection on semen samples[18]. All patients gave immunological study-specific written informed consent covering sampling, storage, and use of biological samples. Healthy volunteers enrolled in the PREVAC (Partnership for Research on Ebola Vaccination) vaccine trial Guinean center agreeing to participate in the immunological evaluation were included, at baseline, as controls. Whole-blood, PBMCs, and serum samples were collected and stored on site. The study protocols were approved by the Research Committee of the National Ebola Response Coordination and the National Ethics and Health Research Committee in Guinea and ethics committees in France (INSERM/CEEI, IRD/CCDE).

**Blood EBOV-RNA and serum EBOV antibody analysis**. The Real-Star Filovirus Screen RT-PCR Kit 1.0 (Altona Diagnostics GmbH, Hamburg, Germany) was used to test for EBOV-RNA at the European West African Mobile Laboratory (EuWamlab) or the Institut National de Sante Publique (INSP) in Conakry. EBOV-specific IgG was quantified in Luminex assays, as previously described[44]. Samples were considered positive for IgG against EBOV if they reacted simultaneously and repeatedly with NP and glycoprotein (GP) or NP ± viral protein (VP40) + GP (weakly positive: 200–399 MFI)[45].

**Quantification of serum analytes**. Twenty-nine parameters were analyzed in serum samples, by magnetic bead assays or ELISA: Human XL Cyt Disc Premixed Mag Luminex Perf Assay Kit (CCL2/MCP-1, CCL4/MIP-1β, RANTES/CCL5, CCL11/Eotaxin, CCL19/MIP-3β, CCL20/MIP-3α, CD40Ligand, Fractalkine/CX3CL1, CXCL10/IP-10, EGF, Flt-3 Ligand, granzyme B, IL-1RA, IL-6, IL-7, IL-8/CXCL8, IL-10, IL-15, PD-L1/B7-H1, TNF-α, TRAIL/TNFSF10, and VEGF) (R&D Systems, #FCSTM18-22). For the Human Magnetic Luminex Assay (CD163, ST2, and CD14, LBP) (R&D Systems, LXSAHM-02), plates were prepared according to the manufacturer's recommendations and read on a Bio-Plex 200 System™ version

6.1. Amphiregulin (#DAR-00), FABP2/IFABP (#DFBP20), and haptoglobin (#DHAPGO) were determined with Human Quantikine ELISA Kits (R&D Systems), according to the manufacturer's instructions.

**Cell phenotyping**. Immune phenotyping was performed with an LSRII Fortessa 4-laser (488, 640, 561, and 405 nm) flow cytometer (BD Biosciences), and Diva software version 6.2. FlowJo software version 9.9.6 (Tree Star Inc.) was used for data analysis. Multiparametric flow cytometry panel was performed using a battery of antibodies: anti-CD38 FITC #340909 (dilution: 1/10), anti-HLA-DR PE #347401 (dilution: 1/10), anti-CD4 BV421 #562424 (dilution: 1/33), anti-CD8 APCH7 #560179 (dilution: 1/20), anti-CD3 Alexa 700 #557943 (dilution: 1/100), anti-CCR7 Alexa 647 #557734 (dilution: 1/33), CD21 PE #555422 (dilution: 1/10), CD27 APC #337169 (dilution: 1/100), CD45 Alexa 700 #560566 (dilution: 1/100), anti-CD56 PECF594 #564849 (dilution: 1/500), anti-HLA-DR BV605 #562845 (dilution: 1/50), anti-CD33 BV421 #562854 (dilution: 1/20), anti-CD141 BV711 #563155 (dilution: 1/20), anti-CD45RA PerCpCy5.5 #563429 (dilution: 1/20), anti-HLA-ABC BV786 #740982 (dilution: 1/50), anti-CD86 PECF594 #562390 (dilution: 1/20) (all from BD Biosciences), anti-CD45RA PEefluor 610 #61-0458-42 (ebiosciences) (dilution: 1/20); anti-CD19 PC7 #IM3628 (Beckman Coulter) (dilution: 1/100), anti-CD38 PercpCy5.5 #303522 (dilution: 1/100), anti-IgM Pacific Blue #314514 (dilution: 1/100), anti-CD71 BV650 #334116 (dilution: 1/20), anti-CD20 APC Cy7 #302314 (dilution: 1/20), anti-CD16 APC Cy7 #302018 (dilution: 1/20), anti-CD14 BV605 #301834 (dilution: 1/20), anti-CD161 BV421 #339914 (dilution: 1/20), anti-NKG2D PercpCy5.5 #320818 (dilution: 1/20), anti-NKp46 PC7 #331916 (dilution: 1/20), anti-CD1c PE-Cy7 #331516 (dilution: 1/50), anti-CD40 PE #334308 (dilution: 1/20), Lineage FITC #348801 (BioLegend) (dilution: 1/100), IgD FITC #H15501 (Invitrogen) (dilution: 1/100), and anti-CD123 APC #130-113-322 (Miltenyi Biotec) (dilution: 1/50). For intracellular cytokine staining (ICS) analyses, cells were stained with surface monoclonal antibodies: anti-CD4 PE PECF594 #562281 (dilution: 1/33), anti-IFNγ FITC #557718 (dilution: 1/20), anti-TNFα PE-Cy7 #557647 (dilution: 1/20), anti-MIP-1β PE #550078 (dilution: 1/200), and anti-IL-2 BV421 #564164 (dilution: 1/20) (all from BD Biosciences). All antibodies were commercially available. See the corresponding manufacturer datasheets on webpages for reference and validation

CD4[+] and CD8[+] T cells were analyzed for CD45RA and CCR7 expression, to identify the naive, memory, and effector cell subsets, and for co-expression of the activation markers HLA-DR and CD38. CD19[+] B cell subsets were analyzed for the CD21 and CD27 markers. Antibody-secreting cells (plasmablasts) were identified as CD19[+] cells expressing CD38 and CD27. We used CD16 and CD56 to identify NK cell subsets. HLA-DR, CD33, CD45RA, CD123, CD141, and CD1c were used to identify DC subsets, as previously described.[46]

**Characterization of EBOV-specific immune responses**. Cellular responses to EBOV peptides were assessed with EpiMax technology[47]. Briefly, PBMCs were stimulated in vitro with 158 overlapping 15-mer peptides (11-amino acid overlaps), covering the Ebola virus Mayinga variant GP, in two pools of 77 (EBOV1) and 81 peptides (EBOV2) (JPT Technologies). Cell functionality was assessed by ICS, with Boolean gating (Fig. 5b, Supplementary Appendix). The flow cytometry panel included a viability marker, CD3, CD4, and CD8 to determine T cell lineage, and CD107a, IFN-γ, TNF, MIP-1β, and IL-2 antibodies. Distributions were plotted with SPICE version 5.22, downloaded from http://exon.niaid.nih.gov/spice[48].

**RNA isolation and messenger RNA (mRNA)-sequencing**. Total RNA was purified from whole blood with the Tempus™ Spin RNA Isolation Kit (Invitrogen) and globin mRNA depletion carried out with human GLOBINclear™ Kit (Invitrogen). Gene expression profiles were analyzed by mRNA-sequencing. RNA was quantified with the Quant-iT RiboGreen RNA Assay Kit (Thermo Fisher Scientific) and quality control was performed on a Bioanalyzer (Agilent). Libraries were prepared with the TruSeq® Stranded mRNA Kit, according to the Illumina protocol. Libraries were sequenced on an Illumina HiSeq 2500 V4 system. The single read sequencing depth was was about 50 million reads, with a fragment length of 101 bp. Sequencing quality control was performed with Sequence Analysis Viewer. FastQ files were generated from.bcl files on Illumina BaseSpace sequence hub. After trimming (QPhred score ≥25), reads were aligned to the hg19 human reference genome, using STAR - 2.5.3ar, and quantified relative to annotation model hg19 - GENCODE Genes - release 19, with Partek E/M (Partek® Flow® software, Copyright ©; 2019 Partek Inc., St. Louis, MO, USA). Counts were normalized by the counts per million (+0.0001) method. PCA was carried out using the Partek software. Differentially expressed genes were identified by using normalized read counts as input for Gene-specific analysis (GSA) performed on Partek® Flow® software. Only genes with adjusted p values (false discovery rate) ≤0.05 and a fold-change in expression ≥1.5 were considered to be differentially expressed. The differentially expressed genes were subjected to functional enrichment analysis with the Ingenuity Pathway software.

**Statistical analysis**. GraphPad Prism software version 8 was used for nonparametric statistics and plots, as described in the figure legends. Correlations between phenotypic, seric, and genetic markers were analyzed by Spearman's rank correlations. A PCA with a Spearman's correlation matrix was performed. Statistical

analyses were carried out with SAS (version 9.3 or higher, SAS Institute, Cary, NC, USA), R (version 3.6, The R Foundation for Statistical Computing, Vienna, Austria), and XLSTAT (version 2011.4.04, Addinsoft, Paris, France). Associations between markers and clinical symptoms have been explored by Wilcoxon's ranksum test and then using random forests to capture nonlinear relationships (data not shown). Final adjusted analyses were performed by using logistic regressions. Fits have been evaluated by Hosmer–Lemeshow tests.

**Reporting summary**. Further information on research design is available in the Nature Research Reporting Summary linked to this article.

## Data availability

RNA-sequencing data that support the findings of this study have been deposited in Gene Expression Omnibus (GEO) repository with the accession codes GSE143549. The hg19 human reference genome can be found in the NCBI database (https://www.ncbi. nlm.nih.gov/assembly/GCF_000001405.13/). Source data are provided with this paper.

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

## Acknowledgements

We thank The French Task Force against Ebola, INSERM, and IRD for institutional support, and the Postebogui team for their daily work and the survivors. We also thank Aminata Diallo, head of the immunology laboratory at Institut National de Santé Publique, and Nathan Peiffer-Smadja and Maxime Schvartz for their help in setting up a laboratory in Conakry. This work was supported by INSERM and by the Investissements d'Avenir program, Vaccine Research Institute (VRI), managed by the ANR under reference ANR-10-LABX-77-01.

## Author contributions

Y.L., C.L., and A.W. conceived and designed the study. A.K.K., S.M., J.-C.F., A.T., H.R., E.D., C.L.-M., and L.K. participated in sample collection. E.F., C.L.e., S.F., B.P.H., M.D., M.K., and A.A. performed experiments and analyzed data. A.W., Y.L., H.H., C.L.e., R.T., and A.A. analyzed and interpreted data. A.W. and Y.L. drafted the first version and wrote the final version of the manuscript. All authors approved the final version.

## Competing interests

The authors declare no competing interests.

## Additional information

## PostEboGui Study Group

Ahidjo Ayouba[3], Eric Delaporte[3], Alpha Kabinet Keita[3,4], Lamine Koivogui[6], Christine Lacabaratz[1], Claire Levy Marchal[8], Yves Levy[1,10], Hervé Raoul[5] & Abdoulaye Touré[3,4,6]

