## [Peer Review File · Nature Communications]

Reviewers' comments:

Reviewer #1, expert in hemorrhagic virus pathogenesis (Remarks to the Author):

The manuscript entitled "Long-lasting severe immune dysfunction in Ebola virus disease survivors" by Wiedemann et al. assesses the immune profile and gene expression profile of 35 Ebola survivors from the West African Ebola epidemic an average of almost two years after discharge from treatment Centers in Guinea. The authors report increases in a number of biomarkers of inflammation and immune activation in this group of survivors. Interestingly, the authors suggest that there may be substantial long-term gut damage as reported for other chronic infectious diseases in the Ebola survivors. Overall, the study is well done and important given the rarity of human data from Ebola patients. The authors note that they did not perform gut biopsies on these survivors. However, that data would have strengthened the paper in terms of making a definitive connection to long-term gut damage. The paper could be improved by addressing the following comments:

1. How does the metadata correspond with cytokine levels? For example, are patients that experience joint pain, fatigue, or ocular abnormalities associated with higher levels of inflammation within this dataset?
2. Was there any evidence of persistent infection in immune privileged sites of the survivors, e.g., semen of male survivors? Localized biosamples from patients with ocular disorders?

Reviewer #2, clinical expert on Ebola virus infection (Remarks to the Author):

In this paper, the authors analysed the inflammatory profile and the immune signature in 35 Guinean EVD survivors from the last West African outbreak after a median of 23 months post discharge. They found a persistent high level of inflammatory cytokines, markers of immune activation and of gut tissue damage. Moreover, a higher frequency of CD8 T, exhausted B cells and non-classical NK cells was found when compared to healthy donors. Interestingly, all survivors presented a persistent and robust EBOV specific B and T cell response. Finally, deep sequencing analyses of gene expression in the blood showed a significant enrichment in the expression of genes associated with IFN signalling, complement, pattern recognition receptor and acute-phase response signalling.

Overall, the manuscript is clear and technically sound and presents evidences of unexpected long lasting persistence of inflammatory soluble mediators, immune activation and gut tissue damage marker in Ebola survivors after a really long term follow-up. Moreover, the observation of a very high frequency of EBOV specific T cell response in the absence of reported viral replication highlights the possible persistence of viral protein and/or virus in immune sanctuary that may drive the survival of such a high frequency of specific cell clones.

Major issues:

- In Table 1, the authors provided a list of post-EVD symptoms in the enrolled survivors. The persistence of an inflammatory immune profile after 2 years from the acute phase is intriguing and may drive the occurrence and severity of clinical sequelae. Nevertheless, no association between immune parameters and clinical post-EVD symptoms has been performed. Is there any preferential associations between specific immune markers and specific symptoms? Did the survivors with mild or no EVD sequelae present lower inflammatory profile or a different immune signature?
- The persistent high level of markers associated with the loss of gut barrier integrity highlights the role of microbial translocation in maintaining systemic inflammation in EBOV survivors. Did the author find any correlation among I-FABP, LBP, sCD14 and other inflammatory mediators.
- The immune dysfunction/impairment associated to Ebola survivors has been well characterized and has been also associated to the reactivation of latent herpesvirus (e.g., Epstein Barr Virus, Agrati et al., Cell Death and Dis 2016). Did the author have any data about possible herpesvirus reactivations that could contribute to the immune activation persistence?

Reviewer #3, expert in human immunophenotyping (Remarks to the Author):

Summary

Long-term clinical sequelae are reported in Ebola virus disease (EVD) survivors (EBOV_S), including both mental and physical health issues. However, the persistent influence from EVD on the immune system and the long-term immune profiles after Ebola virus infection are unclear. This paper presents a comprehensive measurement of immune status in EBOV_S, including measurements on serum proteins, immune cell phenotypes, and whole blood gene expression. Based on comparison with healthy controls from the same geographic area, the authors derived immune parameters significantly different between EBOV_S and healthy. This study provides a nice data set to explore blood based immune signatures potentially associated with EBOV_S. However, the analysis performed is quite rudimentary and the biological insights drawn are limited.

Major comments

1. The authors claimed that the signatures uncovered largely reflect persistence of immune response to EVD and/or chronic inflammatory activation reminiscent of other inflammatory diseases. However, besides highlighting some pathways that may be involved, no comparative analysis was made with such signatures. For example, given previous data sets on immune responses during acute states of EVD, the author can compare their signatures with those, as well as with other chronic inflammatory diseases they highlighted in a rigorous manner.
2. Since this study examines 2 years post EVD, it is important to document to the best extent possible what happened to these individuals during those 2 years – were there other infections and what about their vaccination status? These data are also useful for assessing correlates of clinical outcomes (see below.)
3. How well matched is the healthy cohort to the EBOV_S subjects assessed here, e.g., age and sex, and infection and vaccination history? For example, if the EBOV_S subjects were more susceptible to other infections post EVD, one could imagine that the signatures uncovered here are secondary to EVD and should not be interpreted as persistent immune signatures.
4. A strength of this study is that it starts to provide data to address the "need for resources for long term follow-up of survivors". However, the authors claimed "a set of biological and genetic markers could be used to define a signature of 'Chronic Ebola virus diseases(CEVD)' " and "for assessing clinical outcomes". There is insufficient information provided that would enable readers to see how these markers are related to clinical outcomes. The broader implications of the signatures should be further explored . For example, a scoring system could be designed with the blood protein and gene signatures to see if they could successfully predict clinical CEVD. Also, since the time to infection can be variable, this information should be incorporated in such analyses. I'm also curious whether these signatures are associated with the time of discharge or infection?
5. All the comparisons in Figures 1-3 are bulk level between EBOV_S and healthy donors(HD). However, along the lines of the point above, the variation of each individual within the group could be large. Correlating the parameters of each subject with the health condition and outcomes will provide further evidence to assess the biological relevance of these signatures. In addition, more detailed information about the clinical phenotypes of EBOV_S health in general, could be provided as they would be helpful to assess the phenotypic heterogeneity of this cohort.
6. It's a strength to collect multi-modal data types, however, the connection among the different types of signatures is not clear. For example, do levels of certain markers (serum proteins, cellular phenotypes) agree with blood transcriptomic status in each subject? Does the subject with higher expression of IFN signaling related genes have a higher level of CD40 in DCs? Such correlative analyses among parameters could provide more biological insights and one can also start to evaluate the relative contributions of the markers to outcomes. While the number of subjects is relatively small, if the clinical outcome is well defined, there may be power for such analyses.
7. Figures 1 and 2 only provided some of the cytokines and cell populations, while the Results part

is purely descriptive. Since the gene expression data indicate enrichment of INF related genes in the EBOV_S group, what about different IFN levels as pro-inflammatory cytokines? What about other innate cells like monocytes, neutrophils and macrophages in blood? Otherwise, the authors should provide more context in the Results section to justify why these cytokines and cell populations are representative to highlight. In general, all results and raw data tables should be made available and deposited in such a resource study.

Minor comments

1. Figure 3 only provides the cytokine response of EBOV_S responses to peptides. It would be helpful to provide the stimulation data of the HD group so one can get a better sense of the specificity of the T cell response and its polyfunctional cytokine production profile.
2. The column label is missing in figure 4C (which should be individual subjects).

NCOMMS-19-29939; Wiedemann et al.
Authors' response to reviewers

Reviewer # 1

The manuscript entitled "Long-lasting severe immune dysfunction in Ebola virus disease survivors" by Wiedemann et al. assesses the immune profile and gene expression profile of 35 Ebola survivors from the West African Ebola epidemic an average of almost two years after discharge from treatment Centers in Guinea. The authors report increases in a number of biomarkers of inflammation and immune activation in this group of survivors. Interestingly, the authors suggest that there may be substantial long-term gut damage as reported for other chronic infectious diseases in the Ebola survivors. Overall, the study is well done and important given the rarity of human data from Ebola patients. The authors note that they did not perform gut biopsies on these survivors. However, that data would have strengthened the paper in terms of making a definitive connection to long-term gut damage. The paper could be improved by addressing the following comments:

1. *How does the metadata correspond with cytokine levels? For example, are patients that experience joint pain, fatigue, or ocular abnormalities associated with higher levels of inflammation within this dataset?*

Authors' response: In order to respond to this comment, we performed new analyses looking for associations between cytokine levels, cell population profiles and clinical symptoms in n=34 survivors.

i) regarding clinical symptoms, we reviewed clinical records from 34 patients included in this study at the harvest time point (V0), within one month (M1) and within three months (M3). At V0, 16/34 patients experienced at least one symptom with the following distribution: general symptoms (fatigue, anorexia, fever, pallor, abdominal pain, pelvic pain) (n=14), musculoskeletal symptoms (arthralgia, myalgia) (n=4), neurological symptoms (headache, insomnia, vertigo, sensory disorders) (n=10), ocular symptoms (conjunctivitis, ocular disorders, ocular pain) (n=2). Within M1, 18/34 experienced at least one symptom. These symptoms were general signs (n=16), musculoskeletal (n=7), neurological (n=12) and ocular (n=2). Within M3, 27/34 patients experienced at least one symptom. More precisely, n=23 patients experienced a general symptom, n=13 patients a musculoskeletal symptom, n=20 patients a neurological symptom and n=3 patients an ocular symptom.

ii) the analysis of the distribution of immunological markers according to the presence of clinical symptoms was performed using Wilcoxon rank sum statistical test (without test multiplicity correction in this explanatory analysis). At V0, we found that survivors with symptoms exhibited **a lower frequency of blood pDC** as compared to those without symptoms (median 0.2% IQR [0.16-0.25] vs 0.38% [0.26-0.43]; $P=0.005$). A trend ($P=0.06$) for an association with a higher frequency of the **non-classical CD56⁻CD16⁺NK** cell population (16.1% [9.32-23.12] vs 9% [3.7-14.2]) population was noted. These associations at V0 were still significant within M1, where individuals with symptoms were those with low and high frequencies of blood **pDC** (0.2% [0.15-0.25] vs. 0.38% [0.28-0.42]; $P=0.007$) and blood **CD56⁻CD16⁺ NK** cell (16.1% [9.07-23.17]) vs. 8.43% [3.4-13.8]; $P=0.02$) populations, respectively. A trend ($P=0.04$) to a

decrease of **sCD163** (583.4 ng/ml [375.44-836.45] vs. 880.06 ng/ml [595.34-1182.4]) levels was noted in individuals with symptoms within M1.

A logistic regression model of the presence of symptoms adjusted for the three above markers showed a significant independent effect of each of these markers (all P values <0.02) with a reasonable goodness of fit statistics (Hosmer and Lemeshow test: $P=0.11$).

iii) Additional analyses were performed to look at non-linear relationships using random forests and also for any association with gene expression but no significant additional association was found.

Globally, these new analyses revealed a low frequency of pDC contrasting with an expansion of non-classical NK cell populations in the blood of individuals with symptoms pointing out a persistence of innate immunity dysfunction. These new data are now reported in the results section (page 6 lines 95-108 and pages 10-11 lines 210-227) of the revised manuscript.

2. *Was there any evidence of persistent infection in immune privileged sites of the survivors, e.g., semen of male survivors? Localized biosamples from patients with ocular disorders?*

Authors' response: In the Postebogui cohort, RT-PCR on several body fluids (semen, urine, cervicovaginal secretions, breast milk, and saliva) was performed using the standard RealStarFilovirus Screen RT-PCR Kit 1.0 as well as an additional in-house technique targeting the viral nucleoprotein (Etard *et al.*, The Lancet Infectious Diseases, 2017). Ebola Xpert assay (a higher sensitive assay than PCR) was also used to re-evaluate, retrospectively on available left-over samples, RNA detection on semen samples (Keita *et al.*, OFID, 2019). Among the thirty-five patients enrolled in our study, only one was found positive in the semen at the inclusion in the Postebogui cohort. A new sample of this patient was tested negative 3 months later (at the time of immunological study inclusion, V0). GeneXpert assay on remaining 26 out of 83 semen samples from the nineteen males enrolled in this study did not identify additional positive samples.

Follow-up visit of Postebogui patients' also included eye examination. The eye exam included a measurement of best corrected visual acuity before and after pupil dilation. Intra-ocular samples were not collected in Postebogui patients since this sampling was not approved by the ethics committees (Hereth Hébert *et al.*, American journal of ophthalmology, 2017).

The absence of EBOV detection after re-evaluation (using Ebola Xpert assay) of body fluids from individuals enrolled in this study is now clearly precised in the revised version of the manuscript (page 16 lines 333-337 and page 6 lines 90-92, Material and Methods and Results sections).

Reviewer # 2

In this paper, the authors analyzed the inflammatory profile and the immune signature in 35 Guinean EVD survivors from the last West African outbreak after a

median of 23 months post discharge. They found a persistent high level of inflammatory cytokines, markers of immune activation and of gut tissue damage. Moreover, a higher frequency of CD8 T, exhausted B cells and non-classical NK cells was found when compared to healthy donors. Interestingly, all survivors presented a persistent and robust EBOV specific B and T cell response. Finally, deep sequencing analyses of gene expression in the blood showed a significant enrichment in the expression of genes associated with IFN signalling, complement, pattern recognition receptor and acute-phase response signalling. Overall, the manuscript is clear and technically sound and presents evidences of unexpected long-lasting persistence of inflammatory soluble mediators, immune activation and gut tissue damage marker in Ebola survivors after a really long-term follow-up. Moreover, the observation of a very high frequency of EBOV specific T cell response in the absence of reported viral replication highlights the possible persistence of viral protein and/or virus in immune sanctuary that may drive the survival of such a high frequency of specific cell clones.

Major issues:

1. *In Table 1, the authors provided a list of post-EVD symptoms in the enrolled survivors. The persistence of an inflammatory immune profile after 2 years from the acute phase is intriguing and may drive the occurrence and severity of clinical sequelae. Nevertheless, no association between immune parameters and clinical post-EVD symptoms has been performed. Is there any preferential associations between specific immune markers and specific symptoms? Did the survivors with mild or no EVD sequelae present lower inflammatory profile or a different immune signature?*

Authors' response: We have performed new analyses to answer to this question (please refer to Reviewer#1 answers, Question 1). Globally, these new analyses revealed a low frequency of pDC contrasting with an expansion of non-classical NK cell populations in the blood of survivors with symptoms pointing out a persistence of innate immunity dysfunction. These new data are now reported in the Results section (page 6 lines 95-108 and pages 10-11 lines 210-227) of the revised manuscript.

2. *The persistent high level of markers associated with the loss of gut barrier integrity highlights the role of microbial translocation in maintaining systemic inflammation in EBOV survivors. Did the author find any correlation among I-FABP, LBP, sCD14 and other inflammatory mediators*

Authors' response: To answer to this question, we have performed new analyses and established a spearman correlation matrix. We show now that sCD14 was positively correlated with several inflammatory mediators such as IL-8, sCD40L, sCD163, LBP, exhausted B cells, CD56⁻CD16⁺ NK cells and negatively correlated with total DC and pDC cells. These correlations were significant at the 0.05 level without test multiplicity adjustment.

A new figure 5C and a sentence is now added in the result section (page 10 lines 200-205) indicating these significant correlations.

3. The immune dysfunction/impairment associated to Ebola survivors has been well characterized and has been also associated to the reactivation of latent herpesvirus (e.g., Epstein Barr Virus, Agrati et al., Cell Death and Dis 2016). Did the author have any data about possible herpesvirus reactivations that could contribute to the immune activation persistence?

Authors' response: We thank the reviewer for the reference to this interesting paper. To answer to reviewer's comment, we looked at a possible reactivation of latent CMV and EBV viruses in survivors using samples collected at the time point of the immunological analysis. We performed EBV PCR and CMV PCR using *artus* EBV RG PCR and *artus* CMV RG PCR kits from Qiagen to amplify DNA on remaining serum from HD and EBOV_S. Threshold quantification kits were 1000 copies/ml and 200 copies/ml for EBV and CMV, respectively. On the samples from 38 HD and 34 EBOV_S analyzed, 100% had negative results on CMV PCR and EBV PCR. One sentence precisising the absence of herpes virus reactivation is now added in the discussion section (page 13, lines 278-279).

To go further, we looked at laboratory tests performed in survivors enrolled in the Postebogui cohort at inclusion and during follow-up visits. We collected C-Reactive Protein (CRP) measurements at the time point of the immunological sampling (V0) and two previous visits: 3 months (V-1) and 6 months (V-2) before V0. CRP medians were 3 mg/ml IQR [2;4], 3 mg/ml IQR [3;4], 3 mg/ml IQR [2;4.5] for V0, V-1 and V-2; respectively. Therefore, no evidence of inflammatory syndrome was found. Figure below for the reviewer only. One sentence is now added in the discussion section (page 13, lines 275-278).

Reviewer #3, expert in human immunophenotyping (Remarks to the Author):

Long-term clinical sequelae are reported in Ebola virus disease (EVD) survivors (EBOV_S), including both mental and physical health issues. However, the persistent influence from EVD on the immune system and the long-term immune profiles after Ebola virus infection are unclear. This paper presents a comprehensive measurement of immune status in EBOV_S, including measurements on serum proteins, immune cell phenotypes, and whole blood gene expression. Based on comparison with healthy controls from the same geographic area, the authors derived immune parameters significantly different between EBOV_S and healthy. This study provides a nice data set to explore blood based immune signatures potentially associated with EBOV_S. However, the analysis performed is quite rudimentary and the biological insights drawn are limited.

Major comments

1. *The authors claimed that the signatures uncovered largely reflect persistence of immune response to EVD and/or chronic inflammatory activation reminiscent of other inflammatory diseases. However, besides highlighting some pathways that may be involved, no comparative analysis was made with such signatures. For example, given previous data sets on immune responses during acute states of EVD, the author can compare their signatures with those, as well as with other chronic inflammatory diseases they highlighted in a rigorous manner.*

Authors' response: We thank the reviewer for this suggestion.

i) As a first part of the response, we would like to point out that following requests from reviewers 1 and 2, we have performed new analyses of the associations between immunological markers and clinical symptoms which underscore the relevance of these pathways in survivors enrolled in this study (Please refer to Reviewer #1 answers, Question 1).

Next, to better answer to reviewer's comment, we performed several comparisons with previous published set of data. However, we would like to point out that these comparisons should be taken with cautious since the technical aspects and analyses could differ across studies.

ii) Regarding transcriptomic signatures observed in EBOV survivors: we have performed two new analyses: first a comparison with a chronic "non-infectious" inflammatory disease such as Systemic Lupus Erythematosus (SLE) and, second with data from a previous cohort of acutely Ebola infected individuals.

a. Comparison of gene expression pathways of survivors with signature reported in SLE patients using Chaussabel's modules (Chaussabel *et al*, Immunity 2008) showed that some relevant modules were similarly differentially expressed in SLE and Survivors such as M3.1 Interferon pathway, M2.1 cytotoxic cells, M2.8 T cells and M2.6 Myeloid lineage. Not unexpectedly, there are significant differences in other pathways such as for example M2.4 (Ribosomal proteins). These new data are represented in the results section (page 11 lines 233-238) and in the new Figure 6 A and B in the revised manuscript.

b. As suggested by the reviewer, we also compared our results with published data on gene expression in Ebola acutely infected individuals (Liu *et al.*, Genome Biology, 2017). The full head to head comparison with these data was not possible because of

more stringent criteria used to define the EBOV acute signature. However, comparison of gene expression pathways of survivors with signature reported in acute EBOV patients showed that the interferon module M3.1, which is overexpressed in acute patients, persists in EBOV_S 2 years after healing. These new data are represented in the results section (page 11 lines 238-244) and in the new Supplementary Table 1 in the revised manuscript.

3. *Since this study examines 2 years post EVD, it is important to document to the best extent possible what happened to these individuals during those 2 years – were there other infections and what about their vaccination status? These data are also useful for assessing correlates of clinical outcomes (see below.)*

Authors' response: Patients enrolled in the Postebogui cohort were followed-up for two years with regular visits and clinical records (Etard *et al.*, Lancet ID, 2017; Keita *et al.*, OFID, 2019). During this follow-up period:

- i) no vaccinations were declared or recorded in clinical forms;
- ii) To evaluate possible other infections or reactivations during this period we measured CMV and EBV viral loads. No reactivation of herpes virus infections was noted (see response to reviewer 2);
- iii) C-Reactive Protein levels assessed at the time of immunological study and in previous cohort visits were unchanged and within normal ranges (see data in response to reviewer 2). Therefore, these clinical and biological data likely suggest no evidence of intercurrent disease which may explain the persistence of an activation/inflammation pathways in survivors enrolled in our immunological study.

We have added sentences in the discussion section (page 13, lines 275-279)

4. *How well matched is the healthy cohort to the EBOV_S subjects assessed here, e.g., age and sex, and infection and vaccination history? For example, if the EBOV_S subjects were more susceptible to other infections post EVD, one could imagine that the signatures uncovered here are secondary to EVD and should not be interpreted as persistent immune signatures.*

Authors' response: Age and sex of the two cohorts are presented in the Table 1. To address reviewer comment, we performed statistical analysis on these two parameters. Median age was not statistically different between the two groups (P=0.2). Healthy donors were predominantly men as compared to EBOV_S (P=0.03). EBOV_S declared no vaccination during follow-up period of this study. CMV and EBV viral load in EBOV_S and HD were all negative in the two groups. CRP measurement in EBOV_S showed no evidence of inflammatory syndrome (please refer to response above).

Healthy donors are volunteers enrolled in the PREVAC clinical trial. Criteria inclusion were: no history of EVD, HIV negative status, no vaccination against Ebola, no vaccination in the past 28 days, no clinically significant acute/chronic condition (based on the judgement of the clinician).

5. *A strength of this study is that it starts to provide data to address the "need for resources for long term follow-up of survivors". However, the authors claimed "a set of biological and genetic markers could be used to define a signature of*

'Chronic Ebola virus diseases (CEVD)' " and "for assessing clinical outcomes". There is insufficient information provided that would enable readers to see how these markers are related to clinical outcomes. The broader implications of the signatures should be further explored. For example, a scoring system could be designed with the blood protein and gene signatures to see if they could successfully predict clinical CEVD. Also, since the time to infection can be variable, this information should be incorporated in such analyses. I'm also curious whether these signatures are associated with the time of discharge or infection?

Authors' response: We thank the reviewer for this comment. We have performed new analyses described in response to Reviewer 1 (please refer to Reviewer #1 answers, Question 1 ii)). Moreover, a new figure describing a spearman correlation matrix is now added in the manuscript (new figure 5C) (please refer to Reviewer #2 answers, Question 2).

With respect to the reviewer, we think that a scoring system may be difficult to set up, at first due to the variability of the techniques and the small number of patients included in this study. Moreover, the robustness of a scoring system would need a validation in a larger number of individuals which is difficult to achieve for practical reasons.

However, for better answering to the reviewer's comment, we have performed a **new integrated analysis** including in the model: i) first, biological markers and clinical symptoms (thanks to the 3 reviewers for this convergent comments), which revealed relevant markers characterizing survivors suffering from the persistence of clinically relevant innate immunity dysfunction (see above). In addition, as revealed by the spearman correlation matrix (new figure 5C), the negative correlation between sCD14 and low pDC in survivors with symptoms seems also coherent with the persistence of a microbial translocation from a gut damage in survivors (Please refer to Reviewer#2, Question 2); ii) Second, the integrated analysis was completed with the projection of transcriptomic data (refer to PCA analysis as new Figure 5D).

Globally, we think that the new integrative analysis provides a global overview of markers that reinforce the specific profile of EBOV survivors.

The inclusion in the PostEbogui cohort started at the time of discharge from the Ebola Treatment Center. For practical reason, and the emergency care of the patients after the diagnostic of Ebola infection, the time to infection was not recorded reliably. Therefore we were not able to include this parameter to the statistical analysis.

- 6. All the comparisons in Figures 1-3 are bulk level between EBOV_S and healthy donors(HD). However, along the lines of the point above, the variation of each individual within the group could be large. Correlating the parameters of each subject with the health condition and outcomes will provide further evidence to assess the biological relevance of these signatures. In addition, more detailed information about the clinical phenotypes of EBOV_S health in general, could be provided as they would be helpful to assess the phenotypic heterogeneity of this cohort.*

Authors' response: Here again, analyses of associations between biological markers and clinical symptoms have been added to the manuscript and are also described in response to the Reviewer 1, Question 1 and above.

7. *It's a strength to collect multi-modal data types, however, the connection among the different types of signatures is not clear. For example, do levels of certain markers (serum proteins, cellular phenotypes) agree with blood transcriptomic status in each subject? Does the subject with higher expression of IFN signaling related genes have a higher level of CD40 in DCs? Such correlative analyses among parameters could provide more biological insights and one can also start to evaluate the relative contributions of the markers to outcomes. While the number of subjects is relatively small, if the clinical outcome is well defined, there may be power for such analyses.*

Authors' response: To better answer to reviewer's comment we have performed a large number of new analyses. The association between the various markers has been explored through several approaches: i) a spearman correlation matrix (shown in response to reviewer 2 question 2 and above, new figure 5C); ii) a Principal Component Analysis (PCA): first including phenotypic and seric markers (new Figures 5A and B) and second with the projection of genes highlighted in our previous analyses (new Figure 5D); iii) several other analyses have been performed to deeply explored relationships between all variables: PCA including all genes, sparse Partial Least Square to explain immunological marker variations according to gene expression. These later analyses are not presented because they did not add more knowledge.

The PCA including phenotypic and seric markers (new Figure 5A) shows that the first two components explained 35% of the variability of the markers. Interestingly, when representing the distribution of the individuals according to their clinical status (EBOV survivors or controls), the first component separated quite well the two groups although this analysis was unsupervised since the clinical status information was not used to construct the components. When looking at the contribution of markers over the two components (new Figure 5B), clearly, the first component separated total DC and pDC (projected on the left) from inflammatory markers (projected on the right). These results are consistent and extended data from the spearman correlation matrix (please refer to Reviewer# 1, Question 1 and Reviewer#2, Question 2 answers; new figure 5C)

A projection of the selected genes differentially expressed between survivors and controls on the same plan (new Figure 5D) shows that all interferon signaling genes consistently aggregated with the inflammatory markers whereas the down-regulated genes *C1S*, *FGFR4* and *MASP2* appeared on the total pDC side.

7. *Figures 1 and 2 only provided some of the cytokines and cell populations, while the Results part is purely descriptive. Since the gene expression data indicate enrichment of INF related genes in the EBOV_S group, what about different IFN levels as pro-inflammatory cytokines?*

Authors' response: Thanks to the reviewer for this comment. To try to answer to the reviewer, we measured IFN- γ , IFN- α , and IFN- β serum levels using Luminex technology. These three cytokines were detected in only 10%, 5% and 0% of the EBOV survivors and lower limit of quantifications were 6.42 pg/ml, 1.63 pg/ml and 1.03 pg/ml, respectively. Therefore, the RNA sequencing data showing an activation of IFN pathways was not associated with significant cytokine levels detectable in the serum from survivors (or at very low concentrations below the detectability threshold) using Luminex technology.

8. *What about other innate cells like monocytes, neutrophils and macrophages in blood? Otherwise, the authors should provide more context in the Results section to justify why these cytokines and cell populations are representative to highlight. In general, all results and raw data tables should be made available and deposited in such a resource study.*

Authors' response: We did not assess neutrophil phenotypic characterization in our samples since our analyses were performed on PBMC isolated from blood samples. Similarly, characterization of macrophage populations is not suitable in blood samples and we did not perform tissue or broncho-alveolar sampling. However, as suggested by the reviewer, we evaluated frequency of blood monocytes using CD14⁺ and CD16⁺ markers. New data showed that pro-inflammatory intermediate monocytes (CD14⁺⁺CD16⁺), were significantly increased in EBOV_S as compared to HD (p=0.003). This new result is now added as new figure 2E.

As suggested by the reviewer, we added more context in the results section to justify cytokines and cell populations highlighted in our study (page 7 lines 115-116, 126-17; page 8 lines 142; 146; 158-162).

Raw data tables were uploaded as a Source data file. RNA sequencing data have been uploaded in the Genome Expression Omnibus (GEO) repository. The source data underlying figures are provided as a Source Data File. This information was added in the revised manuscript in the section "Data availability".

Minor comments

1. *Figure 3 only provides the cytokine response of EBOV_S responses to peptides. It would be helpful to provide the stimulation data of the HD group so one can get a better sense of the specificity of the T cell response and its polyfunctional cytokine production profile.*

Authors' response: To answer to reviewer comments, we evaluated EBOV specific responses in 7 HD with PBMC available in the same conditions as used for EBOV_S. We did not detect specific responses in culture conditions with EBOV peptide pools. No statistical differences in the median frequency of CD4⁺ and CD8⁺ T cells in comparison to control culture conditions (non-stimulated) were found. EBOV-specific CD8⁺ T cell expressing the cytotoxicity markers CD107a and IFN- γ were not significantly detected after stimulation of PBMC from controls with EBOV peptides. These results are now presented in new supplementary Figure 5.

2. *The column label is missing in figure 4C (which should be individual subjects).*

Authors' response: We have added the column label in Figure 4 legend.

Reviewers' comments:

Reviewer #1 (Remarks to the Author):

The authors have adequately addressed my comments.

Reviewer #3 (Remarks to the Author):

This paper presents supporting evidence for long-term, potentially persistent immune effects associated with Ebola infection. In this revision the authors provided more clinical details about the EVD survivors (EBOV_S) – including symptoms, vaccination history and the possibility of other infections. Analyses on the correlations of different immune markers and symptoms are also included, but some of the critical issues we brought up earlier were still not clear and thus the paper could be further improved by addressing the following issues:

The healthy cohort and the EBOV_S cohort are not very well matched in terms of age and sex (according to the p values reported between the two groups). The authors didn't supporting data on why this difference will not influence their analyses. In addition, the vaccination history of healthy donors was only evaluated as whether the individual had vaccination in the past 28 days, while the EBOV_S were assessed 18-25 months after discharge from the treatment center. It is more appropriate to include the vaccination history of healthy donors in their analyses.

The correlation analyses added focused on group level differences between HD and EBOV_S. However, many of the parameters evaluated (e.g. cytokine levels, serum proteins, cell populations) and clinical symptoms were quite heterogeneous EBOV_S group, which is to be expected and potentially very interesting. It would be good to conduct correlation analyses using individual subjects, e.g., between the clinical symptoms and immune parameters.

It was nice to see associations between biological markers and clinical symptoms being reported. However, the results on the comparison of biological markers between subcohorts with or without clinical symptoms at different time points would be easier to digest if they can be illustrated with graphs (specifically, the results described in page 6 lines 95-108 and pages 10-11 lines 210-227).

Reviewer #3 (Remarks to the Author):

This paper presents supporting evidence for long-term, potentially persistent immune effects associated with Ebola infection. In this revision the authors provided more clinical details about the EVD survivors (EBOV_S) – including symptoms, vaccination history and the possibility of other infections. Analyses on the correlations of different immune markers and symptoms are also included, but some of the critical issues we brought up earlier were still not clear and thus the paper could be further improved by addressing the following issues:

1. *The healthy cohort and the EBOV_S cohort are not very well matched in terms of age and sex (according to the p values reported between the two groups). The authors didn't supporting data on why this difference will not influence their analyses. In addition, the vaccination history of healthy donors was only evaluated as whether the individual had vaccination in the past 28 days, while the EBOV_S were assessed 18-25 months after discharge from the treatment center. It is more appropriate to include the vaccination history of healthy donors in their analyses.*

Authors' response: Median age was not statistically different between the two groups (P=0.2), but in order to answer reviewer's comment we performed a new spearman correlation matrix between serum soluble mediators and cell phenotypes (as Figure 5C in the manuscript) adjusted for age (partial correlation coefficient). Please refer to figure below (figure for reviewer only):

This spearman correlation matrix is nearly identical to the Figure 5C in the manuscript showing that age did not influence these analyses.

Healthy donors were predominantly men as compared to EBOV_S (P=0.03). It was difficult to have samples from healthy donors from Conakry so we took the

opportunity of PREVAC clinical trial. First subjects enrolled in trial were predominantly men (available samples at the time of our study). Unfortunately, it is not possible to perform same analysis as presented above as sex and the sample size does not allow a proper stratification.

Healthy donors are volunteers enrolled in the PREVAC clinical trial in Conakry, Guinea. One criteria inclusion was no vaccination in the past 28 days. Guinea is a country with a weak health system so it is difficult to monitor past vaccinations. Nevertheless, to our knowledge, between the end of EBOV outbreak in Guinea in June 2016 and sampling of these donors in April/May 2017, no massive adult vaccination programs were conducted in Conakry (Alliance for International Medical Action (ALIMA) non-governmental organization, personal communication).

2. The correlation analyses added focused on group level differences between HD and EBOV_S. However, many of the parameters evaluated (e.g. cytokine levels, serum proteins, cell populations) and clinical symptoms were quite heterogeneous EBOV_S group, which is to be expected and potentially very interesting. It would be good to conduct correlation analyses using individual subjects, e.g., between the clinical symptoms and immune parameters.

Authors' response: It is true that there was a substantial heterogeneity that led to some signals in term of association between biological markers (Figure 5, line 192 to 209) and also differences in concentrations according to the presence of clinical symptoms, that is association between markers and clinical symptoms (line 210 to 224). According to the reviewer comment, we clarified the wording to improve the understanding for the clinical audience of what has been done, especially line 221-224 that read as follow: *“The independent association of each of these markers (i.e. pDC, CD56-CD16+ and sCD163) with the presence of symptoms was confirmed by a multivariable logistic regression model (all P values <0.02) with a reasonable goodness of fit statistics (Hosmer and Lemeshow test: P=0.11).”*

3. It was nice to see associations between biological markers and clinical symptoms being reported. However, the results on the comparison of biological markers between subcohorts with or without clinical symptoms at different time points would be easier to digest if they can be illustrated with graphs (specifically, the results described in page 6 lines 95-108 and pages 10-11 lines 210-227).

Authors' response: We thank the reviewer for this suggestion. To address this comment supplementary figure 2 showing distribution of the symptoms at V0, within M1 and within M3 and supplementary figure 7 showing the distribution of immunological markers according to the presence of clinical symptoms were added in the manuscript.

REVIEWERS' COMMENTS:

Reviewer #3 (Remarks to the Author):

The authors have addressed this reviewer's previous comments. For Fig s7, it was informative to see the association between the markers and the "no symptom"/"having any symptoms" category. It could be even more informative to show actual scatter plots of the individual subject/data-points, but if the raw data is available (so interested readers can examine if they like to), this is not an issue. On this note, it would be good to indicate in the Data Availability what specific data sets are available via public repositories or are attached to the paper via the Nature website (is that the "Source Data File"?), as there seems to be some data that would need to be requested from the corresponding author via "reasonable request" (I copied the statement below for clarification.)

"RNA sequencing data that support the findings of this study have been deposited in Gene Expression Omnibus (GEO) repository with the accession codes GSE143549. The source data underlying Figs 1, 2, 3; Table 1 and Supplementary Figs 3, 5 and 6 are provided as a Source Data File. Other data are available from the corresponding author upon reasonable request."

Reviewer #3 (Remarks to the Author):

The authors have addressed this reviewer's previous comments. For Fig s7, it was informative to see the association between the markers and the "no symptom"/"having any symptoms" category. It could be even more informative to show actual scatter plots of the individual subject/data-points, but if the raw data is available (so interested readers can examine if they like to), this is not an issue. On this note, it would be good to indicate in the Data Availability what specific data sets are available via public repositories or are attached to the paper via the Nature website (is that the "Source Data File"?), as there seems to be some data that would need to be requested from the corresponding author via "reasonable request" (I copied the statement below for clarification.)

"RNA sequencing data that support the findings of this study have been deposited in Gene Expression Omnibus (GEO) repository with the accession codes GSE143549. The source data underlying Figs 1, 2, 3; Table 1 and Supplementary Figs 3, 5 and 6 are provided as a Source Data File. Other data are available from the corresponding author upon reasonable request."

Authors' response: We thank the reviewer for these comments. We modified the supplementary figure 7 in order to show individual data points within the box plots. We have deleted the sentence "Other data are available from the corresponding author upon reasonable request" from the Data available section as all the data are now either deposited in a public repository (Figure 4) or provided as a Source data file